# Epithelial-to-Mesenchymal Transition and Phenotypic Marker Evaluation in Human, Canine, and Feline Mammary Gland Tumors

**DOI:** 10.3390/ani13050878

**Published:** 2023-02-28

**Authors:** Alessandro Sammarco, Chiara Gomiero, Giorgia Beffagna, Laura Cavicchioli, Silvia Ferro, Silvia Michieletto, Enrico Orvieto, Marco Patruno, Valentina Zappulli

**Affiliations:** 1Department of Comparative Biomedicine and Food Science, University of Padua, 35020 Padua, Italy; 2Department of Microbiology, Immunology, and Molecular Genetics, University of California, Los Angeles, CA 90095, USA; 3Department of Cardio-Thoraco-Vascular Sciences and Public Health, University of Padua, 35128 Padua, Italy; 4Breast Surgery Unit, Istituto Oncologico Veneto, IRCCS, 35128 Padua, Italy; 5Department of Pathology, Santa Maria della Misericordia Hospital, 45100 Rovigo, Italy

**Keywords:** E-cadherin, epithelial-to-mesenchymal transition, mammary tumors, *SNAIL*, *TWIST*, *ZEB*

## Abstract

**Simple Summary:**

In this study we addressed the analysis of human breast cancer and canine and feline mammary tumors with regard to the expression, at either gene or protein level, of some molecules that are related to the capacity of an epithelial cell to become mesenchymal (epithelial-to-mesenchymal transition), acquiring higher ability to metastasize. In our samples, some typical markers of this transition were not higher at mRNA levels in tumors than in healthy tissues, indicating that some other markers should be investigated. Instead, at protein levels, some molecules such as vimentin and E-cadherin were indeed associated with higher aggressiveness, being potential useful markers. As already described in the literature, we also demonstrated that feline mammary tumors are close to an aggressive subtype of human breast cancer called triple negative, whereas canine mammary tumors are more similar to the less aggressive subtype of human breast cancer that expresses hormonal receptors.

**Abstract:**

Epithelial-to-mesenchymal transition (EMT) is a process by which epithelial cells acquire mesenchymal properties. EMT has been closely associated with cancer cell aggressiveness. The aim of this study was to evaluate the mRNA and protein expression of EMT-associated markers in mammary tumors of humans (HBC), dogs (CMT), and cats (FMT). Real-time qPCR for *SNAIL*, *TWIST*, and *ZEB*, and immunohistochemistry for E-cadherin, vimentin, CD44, estrogen receptor (ER), progesterone receptor (PR), ERBB2, Ki-67, cytokeratin (CK) 8/18, CK5/6, and CK14 were performed. Overall, *SNAIL*, *TWIST*, and *ZEB* mRNA was lower in tumors than in healthy tissues. Vimentin was higher in triple-negative HBC (TNBC) and FMTs than in ER+ HBC and CMTs (*p* < 0.001). Membranous E-cadherin was higher in ER+ than in TNBCs (*p* < 0.001), whereas cytoplasmic E-cadherin was higher in TNBCs when compared with ER+ HBC (*p* < 0.001). A negative correlation between membranous and cytoplasmic E-cadherin was found in all three species. Ki-67 was higher in FMTs than in CMTs (*p* < 0.001), whereas CD44 was higher in CMTs than in FMTs (*p* < 0.001). These results confirmed a potential role of some markers as indicators of EMT, and suggested similarities between ER+ HBC and CMTs, and between TNBC and FMTs.

## 1. Introduction

Mammary gland cancer is the most common tumor in women [1] and in female dogs [2], and the third most common neoplasia in cats [3]. Human breast cancer (HBC) is classified into four main subtypes according to the expression of estrogen receptor (ER), progesterone receptor (PR), and epidermal growth factor receptor ERBB2, as follows: (i) Luminal A tumors (ER+ and/or PR+, ERBB2-); (ii) Luminal B tumors (ER+ and/or PR+, ERBB2+); (iii) ERBB2-overexpressing tumors (ER-, PR-, ERBB2+); and (iv) triple-negative (ER-, PR-, ERBB2-) breast cancer (TNBC) [4]. TNBCs are typically high-grade carcinomas characterized by an aggressive behavior and a poor prognosis, with high risk of distant metastasis and death [5]. Canine mammary tumors (CMTs) are classified based on morphologic features [6]. Fifty per cent of CMTs are malignant with a 20% risk of metastasis [7]. The majority (80–90%) of feline mammary tumors (FMTs) are characterized by a highly aggressive behavior that leads to rapid progression and distant metastasis development [8,9]. Typically, FMTs lack the expression of ER, PR, and ERBB2, and have been considered a remarkable spontaneous model for TNBC [10,11,12,13,14,15,16]. In all three species, mammary tumors exhibit both inter- and intra-tumor heterogeneity as a consequence of genetic and non-genetic aberrations [17].

Over the past 20 years, the investigation of cell differentiation/phenotypic markers has been used in both human and veterinary medicine, primarily to improve our knowledge of the histogenesis of mammary tumors [18]. In the normal human, canine, and feline mammary gland, two cell subpopulations are present: luminal epithelial cells, positive for cytokeratin (CK) 7, CK8, CK18, and CK19; and basal/myoepithelial cells, variably positive for CK5, CK6, CK14, CK17, SMA, calponin, vimentin, and p63 [19]. In HBC, the evaluation of cell differentiation proteins is frequently performed in association with routine diagnostic markers (ER, PR, ERBB2, and Ki-67) to better classify this tumor. The identification of HBC subtypes has a diagnostic, prognostic, and therapeutic value, and is associated with the cell differentiation and epithelial-to-mesenchymal transition (EMT) status of the neoplastic population according to a hierarchical model [20]. 

EMT is a key event that neoplastic epithelial cells use to acquire a mesenchymal phenotype [21]. As a result, tumor cells obtain the ability to detach from the primary tumor mass, invade the surrounding tissue, migrate throughout the body, and eventually give rise to metastases in distant organs [22]. The classical EMT is characterized by a decreased expression of epithelial markers and a complementary upregulation of mesenchymal markers. Classical EMT transcription factors, namely snail family transcription repressor 1/2 (SNAIL), TWIST, and zinc-finger-enhancer binding protein 1/2 (ZEB) are known to orchestrate EMT by regulating cell adhesion, migration, and invasion, also interacting with different signaling pathways and microRNAs [22,23]. Although this is a well-de-scribed process that promotes metastasis formation, accumulating evidence suggests the existence of an intermediate state called partial EMT or hybrid E/M, whereby both epithelial and mesenchymal markers are co-expressed in cancer cells [23,24,25]. 

The aim of this study was to investigate the mRNA expression of classical EMT-related transcription factors *SNAIL*, *TWIST*, and *ZEB* in human, canine, and feline mammary tumors. Additionally, we studied the expression of key proteins involved in the EMT process, including E-cadherin and vimentin, and of proteins related to the tumor phenotype, such as ER, PR, ERBB2, Ki-67, cytokeratin (CK) 8/18, CK5/6, CK14, and CD44.

## 2. Materials and Methods

### 2.1. Tissue Collection

Human samples were collected from the Istituto Oncologico Veneto (IOV, Padua, Italy), whereas canine and feline samples were collected from local veterinary clinics. The human sample collection was approved by the IOV Ethics Committee. All patients or patients’ owners provided informed, written consent to use their samples for this study. Specifically, samples from 5 healthy human mammary gland tissues (MGTs), 5 ER+ HBCs, 5 TNBCs, 4 healthy canine MGTs, 10 canine mammary tumors (CMTs) (5 grade I and 5 grade II), 6 healthy feline MGTs, and 6 grade III FMTs were collected. In this study, to avoid contaminations with other tumor cell subpopulations, we selected only simple tubular carcinomas (STC), which are composed of only one tumor cell subpopulation (luminal epithelial cells) [6]. Healthy MGTs were collected from tumor-bearing patients during the therapeutic/diagnostic surgical procedures, with no additional sampling performed only for the study. Sampling was performed by surgeons. At the time of sampling, most of the tissue was fixed in 4% formaldehyde for histopathology and immunohistochemistry, whereas a peripheral small portion of tumor and normal tissues (approx. 0.5 cm^2^ each) was collected and preserved in RNALater (Ambion, Austin, TX, USA), according to manufacturer’s instructions. In the lab, before RNA extraction, a small portion of each RNALater-preserved sample was fixed in 4% formaldehyde and embedded in paraffin to check the content of the samples themselves. Four-μm tissue sections were stained with hematoxylin and eosin, and slides were visualized under the microscope to further confirm the presence of healthy tissue in the samples labelled as “healthy” and of tumor tissue in the samples labelled as “tumor”.

### 2.2. RNA Extraction and Real-Time Polymerase Chain Reaction

For gene expression analysis, a small portion of each tissue sample preserved in RNALater was used for RNA extraction using Trizol Reagent (Invitrogen, Carlsbad, CA, USA), following the manufacturer’s protocol. The extracted RNA was treated with RNAse-free DNAse I (New England Biolabs, Ipswich, MA, USA). Five-hundred ng of total RNA from each sample was reverse transcribed using the RevertAid First Strand cDNA Synthesis Kit (Invitrogen). The cDNA was then used as a template for quantitative real-time PCR using the ABI 7500 Real-Time PCR System (Applied Biosystem) to evaluate the mRNA expression of the following EMT-related genes: *SNAIL1*, *SNAIL2*, *TWIST1*, *TWIST2*, *ZEB1*, *ZEB2*. All the samples were tested in triplicate. *ACTB* was used as a house-keeping gene. The primer sequences are reported in Table 1. The primers were designed using NCBI Primer-BLAST. To examine primer specificity, the dissociation curves of qPCR products were assessed to confirm a single amplification peak. The qPCR reactions were then purified using the ExoSAP-IT PCR product cleanup (Applied Biosystems) and sequenced at the BMR Genomics (Padua, Italy). The sequences were then verified using the NCBI BLAST database. For data analysis for each sample, the ΔΔCt value was calculated and expressed as a relative fold change (2^−ΔΔCt^), as described in [16]. Real-time PCR efficiency was calculated by performing a dilution series experiment and applying the following formula to the standard curve: efficiency = 10^(−1/slope)^ − 1 [26,27]. Real-time PCR efficiency was between 90 and 100% for all the samples.

### 2.3. Immunohistochemistry

Immunohistochemistry (IHC) was performed on the above-mentioned samples as well as on additional human breast tissue samples from the Division of Anatomic Pathology archive of the University of Padua Hospital, and on additional canine and feline mammary tissue samples from the anatomic pathology archive of the Department of Comparative Biomedicine and Food Science of the University of Padua. Specifically, IHC was per-formed on the following tissue samples: 10 ER+ HBC, 11 TNBCs, 11 CMTs grade I, 11 CMTs grade II, 12 FMTs grade III. Sections (4 μm) were processed with an automatic immunostainer (BenchMark XT, Ventana Medical Systems), as previously described [11]. Briefly, the automated protocol included the following steps: a high-temperature antigen unmasking (CC1 reagent, 60 min), primary antibody incubation (1 h at RT, see below for dilutions), an ultrablock (antibody diluent, 4 min), hematoxylin counterstain (8 min), dehydration, and mounting. Negative controls omitted the primary antibody, whereas adnexa, epidermis, and non-tumor mammary gland, when present, were used as positive controls for CK8/18, CK5/6, CK14, E-cadherin, vimentin, and Ki-67. For ERBB2, an additional technical external positive control was used (ERBB2 3+ HBC), whereas the species-specific cross-reactivity was previously tested in dogs and cats [10,28]. For ER and PR, feline and canine uterus as well as ovary were also stained as positive controls. For CD44, the lymph node was used as positive control. Positive control tissues, typically collected from necropsies, were derived from the same archive as the canine and feline mammary tumor samples. The following antibodies were tested: anti-ER alpha (anti-ERα) (NCL-ER-6F11 1:40, Novocastra in human and feline species—NCL-ER-LH2 1:25, Novocastra in canine species); anti-PR (NCL-PGR-312 1:80, Novocastra in human and feline species); an-ti-ERBB2 (A0485 1:250, Dako in canine and feline species); anti-CK8/18 (NCL-L-5D3 1:30, Novocastra); anti-CK5/6 (D5/16 B4 1:50, Dako); anti-CK14 (NCL-LL 002 1:20, Novocastra); anti-E-cadherin (610182 1:120, BD Biosciences); anti-CD44 (550538 1:100, BD Biosciences); anti-vimentin (M0725 1:150, Dako); and anti-Ki-67 (M7240 1:50, Dako). In the human species, ERBB2 immunolabeling was performed with Bond Oracle HER2 IHC System for BOND-MAX (Leica Biosystems), containing the anti-ERBB2 antibody (clone CB11, ready-to-use). IHC positivity was semi-quantitatively and separately evaluated by ECVP-boarded (V.Z.) and experienced (L.C.) pathologists. Specifically, cytoplasmic and nuclear positivity were measured as a percentage of positive cells for all markers (100 cells per field in 10 high-power fields were counted). ERBB2 was scored as 0, 1+, 2+, and 3+ according to the American Society of Clinical Oncology (ASCO) 2018 recommendations [29] (10% cut-off), with 2+ and 3+ cases considered weakly and strongly positive for complete membrane immunolabeling, respectively. The protein expression of the studied markers was evaluated in the epithelial/luminal component. Additionally, immunolabeling was observed in healthy/hyperplastic adjacent mammary tissue, and in this case normal basal/myoepithelial cells were also evaluated.

### 2.4. Statistical Analysis

Statistical analyses were performed using Prism version 9.3.1 (GraphPad Software, San Diego, CA, USA). To verify mean differences among groups, either the Student’s *t*-test or the one-way ANOVA with Tukey’s multiple comparison test was used, when values were normally distributed. A Mann–Whitney test or Kruskal–Wallis test were used when values were not normally distributed. Normality was tested using the Shapiro–Wilk test. The Spearman’s rank correlation analysis was used to analyze associations between variables. The level of significance was set at *p* < 0.05.

## 3. Results

### 3.1. Gene Expression

We sought to investigate the mRNA expression of the EMT transcription factors *SNAIL*, *TWIST*, and *ZEB* in mammary tumors compared with healthy tissue. In HBC (Figure 1), *SNAIL1* showed a higher mRNA expression in TNBCs when compared with ER+ (*p* < 0.05). Conversely, the mRNA expression of *TWIST1*, *TWIST2*, and *ZEB1* in ER+ and TNBCs was significantly lower than in healthy MGTs (*p* < 0.05). Additionally, TNBCs had a significantly lower mRNA expression of *SNAIL2* and *ZEB2* when compared with healthy MGTs (*p* < 0.05).

In CMTs (Figure 2), *SNAIL1* showed a higher mRNA expression in STC II when compared with healthy MGTs (*p* < 0.01) and STC I (*p* < 0.001). The mRNA expression of *SNAIL2, ZEB1*, and *ZEB2* was lower in tumors than healthy MGTs, although not statistically significant.

In FMTs (Figure 3), tumors showed a lower mRNA expression of *SNAIL1*, *SNAIL2*, *TWIST1*, *TWIST2*, *ZEB1*, and *ZEB2* when compared with healthy MGTs, which was significant only for *ZEB1* (*p* < 0.05). 

### 3.2. Immunohistochemistry

Next, we aimed to study the expression of key proteins involved in the EMT process. The expression of the studied markers was evaluated in the tumor epithelial luminal cell population.

CD44 and ERBB2 staining was membranous, whereas CK8/18, CK5/6, CK14, and vimentin staining was cytoplasmic. E-cadherin staining was present in either or both membrane and cytoplasm and it was separately evaluated. Ki-67, ER, and PR staining was nuclear. As expected, epithelial luminal cells of healthy MGT in all three species were diffusely positive for CK8/18, membranous E-cadherin, ER, PR, and occasionally positive for CK5/6, CK14, and CD44. The basal/myoepithelial cells of healthy MGT in all three species were diffusely positive for CK5/6, CK14, CD44, and vimentin, and occasionally also positive for ER and PR. 

Results for the human, canine, and feline mammary tumors are summarized in Table 2, Appendix A and are graphically represented in Figure 4.

In HBC (Figure 4A), ER+ tumors had a high protein expression (roughly 100%) of CK8/18, whereas they were negative for basal cytokeratins CK5/6 and CK14. In TNBCs, the protein expression of CK8/18, although fairly heterogeneous, was lower than in ER+ (*p* < 0.001) and the protein expression of CK5/6 was higher than in ER+ (*p* < 0.05). In ER+ tumors the protein expression of E-cadherin was predominantly membranous (Figure 5A), whereas in TNBCs E-cadherin protein expression was often lost from the membrane and pre-dominantly cytoplasmic (Figure 5B). Membranous E-cadherin protein expression was higher in ER+ than in TNBCs (*p* < 0.001), whereas cytoplasmic E-cadherin protein ex-pression was higher in TNBCs when compared with ER+ (*p* < 0.001) (Figure 4A). Overall, the expression of this protein was quite heterogeneous across the samples. Interestingly, a strong negative correlation between membranous and cytoplasmic E-cadherin protein expression was found in ER+ (r = −1, *p* < 0.001) (Figure 4B) and in TNBCs (r = −0.9, *p* < 0.001) (Figure 4C). CD44 protein expression was lower in ER+ (Figure 5C) than in TNBCs (Figure 5D), although not statistically significant. Notably, in TNBCs, a strong positive correlation between CK5/6 and CK14 expression (r = 0.8, *p* < 0.01), and a moderate positive correlation between CD44 and vimentin (r = 0.6, *p* = 0.05), were found. 

All CMTs (Figure 4D) were positive (>1%) for ER and, therefore, classified as ER+. ER protein expression was lower in STC II than in STC I (*p* < 0.01). The protein expression of E-cadherin was quite heterogeneous across the samples. As in HBC, a strong negative correlation between membranous and cytoplasmic E-cadherin protein expression was found in the CMTs (r = −0.974, *p* < 0.001) (Figure 4E). In addition, in STC II, a strong positive correlation between CK8/18 and membranous E-cadherin (r = 0.8, *p* < 0.01) and a strong negative correlation between CK8/18 and cytoplasmic E-cadherin (r = −0.8, *p* < 0.01) were found. Interestingly, in STC II, Ki-67 expression was positively correlated with CK8/18 (r = 0.7, *p* < 0.05) and membranous E-cadherin (r = 0.8, *p* < 0.01) expression, and negatively correlated with cytoplasmic E-cadherin expression (r = −0.7, *p* < 0.05). 

All FMTs (Figure 4D) were negative for ER (<1%), PR (<1%), and ERBB2 (either 0 or 1+), and were therefore classified as triple negative. E-cadherin protein expression was quite heterogeneous. As in the HBCs and CMTs, a strong negative correlation between membranous and cytoplasmic E-cadherin protein expression was found (r = −0.984, *p* < 0.001) (Figure 4F). In addition, a strong negative correlation between CK5/6 and vimentin expression was found (r = 0.8, *p* < 0.01). 

CD44 protein expression was higher in the CMTs (Figure 5E) than in the FMTs (*p* < 0.001) (Figure 5F). Vimentin and Ki-67 protein expression was lower in the CMTs than in the FMTs (*p* < 0.001) (Figure 6). 

The expression of the studied markers was not associated with other histopathological features, such as vascular invasion or regional lymph node metastases (data not shown). Moreover, no significant correlations were found between gene and protein expression of the analyzed markers.

## 4. Discussion

In this study, we investigated the expression of genes and proteins involved in one of the processes thought to play a major role in cancer progression: epithelial-to-mesenchymal transition [22]. 

EMT is an evolutionally conserved morphogenetic program during which epithelial cells undergo a series of changes allowing them to acquire a mesenchymal phenotype [21]. During classical EMT, epithelial cells lose the expression of tight junction molecules such as membranous E-cadherin and acquire mesenchymal properties such as migration, invasiveness, and elevated resistance to apoptosis. Transcription factors like SNAIL, TWIST, and ZEB regulate this process and are activated by a variety of signaling pathways, including TGF-α, Notch, and Wnt/β-catenin [30,31,32,33]. 

SNAIL is a classical regulator of EMT that represses E-cadherin transcription in both mouse and human cell lines [34]. In HBC, it has been associated with tumor recurrence and metastasis [35], and with poor patient prognosis [36]. In contrast to the findings of other authors [37], we found that the mRNA expression of *SNAIL2* was significantly lower in TNBCs than in healthy MGTs. In CMTs, *SNAIL1* expression was higher in STC II when compared with healthy MGTs and STC I, indicating a possible association of EMT with a higher aggressiveness of these tumors. *SNAIL2* in CMTs did not show any difference between healthy MGT and tumor tissue, confirming what other authors have also found [38,39,40]. Conversely, in FMTs, there was a trend such that STC III had a lower mRNA expression of *SNAIL1* and *SNAIL2* when compared with healthy MGTs. To the best of our knowledge, SNAIL has never been investigated in feline tumors. 

It is believed that TWIST plays an essential role in cancer metastasis [33]. In HBCs and FMTs, the mRNA expression of *TWIST1* and *TWIST2* was lower in tumors than in healthy MGTs, which differs from what some authors have found in HBC [41], but is similar to what other authors have found in HBC [42] and in FMTs [43]. 

ZEB1 has been implicated in carcinogenesis in breast tissue [44] because it enhances tumor cell migration and invasion [45]. In our samples, *ZEB1* mRNA expression was lower in tumor than in healthy MGTs, as previously reported by other authors in HBC [42]. Although one study examined the expression of *ZEB1* and *ZEB2* in five canine mammary carcinoma cell lines [46], to the best of our knowledge, *ZEB* mRNA expression has never been studied in CMT and FMT tissues. 

Overall, our data suggest that these transcriptional factors are often downregulated in tumors compared with healthy MGTs, except for *SNAIL1* in TNBCs and in CMTs STC II. The RNA isolated from healthy tissues came from the whole mammary gland, which is composed of different cell populations, namely epithelial cells, connective tissue, and fat. Although these transcription factors are barely detectable in normal mesenchymal cells of adult tissues [47], adipose tissue expresses these genes variably [48]. As a result, the mRNA levels of these genes in healthy samples can be dramatically influenced by the presence of non-mammary gland tissues, such as fat.

Moreover, it is possible that the number of cells undergoing classical EMT is low when compared with the tumor bulk, which is known to be characterized by a remarkable intra-tumor heterogeneity [22]. Furthermore, some authors believe that these genes are regulated post-transcriptionally [35,49,50,51]. Furthermore, accumulating evidence suggests the existence of cell populations with a hybrid E/M state, which exhibit increased plasticity and metastatic potential, characterized by the co-expression of epithelial and mesenchymal markers [23,24,25,52]. However, the expression of some of these markers may be associated with a complete EMT status, whereas others may be associated with a partial EMT status. For example, it is believed that SNAIL1 is a stronger inducer of complete EMT than SNAIL2, which is rather associated with a hybrid E/M state [53,54]. This suggests that the choice of the markers to be analyzed is fundamental and may help in identifying intermediate EMT states more precisely. In addition, in order to study the EMT process, it would be interesting in the future to investigate the expression of these markers at a single cell level, using single-cell omics approaches such as Laser Capture Microdissection or single-cell RNA sequencing.

In the present study, we also assessed the protein expression of several phenotypic as well as EMT-related markers, such as ER, PR, ERBB2, CK8/18, CK5/6, CK14, E-cadherin, CD44, vimentin, and Ki-67, in a subset of HBCs, CMTs, and FMTs.

The HBC ER+ samples showed a high expression of luminal CK8/18, and a negative expression of basal CK5/6 and CK14, confirming the strong association between ER+ tumors and highly differentiated glandular cells (CK8/18+), as well as null expression of basal CKs (CK5/6, CK14). In the TNBCs, the protein expression of CK8/18 was highly heterogeneous, whereas the expression of CK5/6 and CK14 was low in most of the samples. This result, in concordance with another study [55], supports the idea that the terms “basal-like cancer” and “triple-negative breast cancer” are not interchangeable. Indeed, only a small percentage of TNBCs are basal-like [56]. The CMTs were positive for ER, whereas the FMTs were negative for ER, PR, and ERBB2. Despite only a few samples being analyzed, these data suggest, as already proposed by other authors [11,57], a similarity between CMTs and HBC ER+ and between FMTs and TNBCs. In CMTs and FMTs, the protein expression of CK8/18, CK5/6, and CK14 was highly heterogeneous, confirming the high inter- and intra-tumor heterogeneity [16,57]. Basal CK14 protein expression was higher in FMTs than in CMTs, confirming that FMTs are more “basal-like” when compared with CMTs [11,12].

E-cadherin is a cellular adhesion molecule, and its disruption may contribute to the enhanced migration and proliferation of tumor cells, leading to invasion and metastasis [58,59,60,61,62]. In our samples, E-cadherin protein expression was evaluated in the membrane and in the cytoplasm of tumor cells, separately. Overall, the expression of E-cadherin was highly heterogeneous across the samples of the three species, confirming once more the high inter-tumor heterogeneity of mammary cancer in the three species. In human ER+ tumors, E-cadherin protein expression was predominantly membranous, whereas in TNBCs it was predominantly cytoplasmic, confirming that the delocalization of the protein is associated with increased tumor aggressiveness [56,63]. These results confirm that it is not only the loss of E-cadherin that correlates with increased tumor aggressiveness, but also the protein translocation from the membrane to the cytoplasm, as already described [64,65,66,67].

Together with E-cadherin, CD44 has been extensively studied in tumor cell differentiation, invasion, and metastasis, and is thought to be involved in the EMT process in HBC [68,69]. Although a few studies on HBC have shown that protein overexpression of CD44 is associated with poor prognosis and metastasis [70], others have shown that downreg-ulation of its expression is correlated with an adverse outcome [68,71]. For this reason, the role of CD44 in the behavior and prognosis of HBC is controversial [71,72]. In our study, CD44 expression was heterogeneous and lower overall in ER+ tumors compared with TNBCs. This trend agrees with study findings by Klingbeil and collaborators, who found high levels of CD44 expression in tumors with a basal-like or triple-negative phenotype, suggesting an association of this protein with an aggressive phenotype in HBC [73]. CD44 was highly expressed (roughly 85%) in our CMT samples, regardless of the tumor grading, as well as in the healthy mammary gland tissues. Moreover, other authors found no differences between benign CMTs, malignant CMTs, and normal mammary gland tissues, suggesting that CD44 is not associated with aggressiveness in canine mammary tumors [74,75,76,77,78]. In FMTs, the expression of CD44 was low overall (approximately 5%). Sarli and collaborators evaluated the intramammary/intratumoral and extramammary/extratumoral expression of CD44 in feline normal mammary tissues, benign tumors, and malignant tumors in relationship to lymphangiogenesis [79]. They found that CD44 had a significantly higher expression in intramammary/intratumor areas compared with extramammary/extratumor areas in both benign and malignant tumors. Additionally, no statistically significant differences in CD44 expression between normal mammary gland, benign tumors, and malignant tumors were found. To the best of our knowledge, no other studies on CD44 expression in FMT tissues are present within the literature. These data, together with our findings, suggest that CD44 is not a useful marker of malignancy in cats.

Another protein that is well-studied and plays a central role in the EMT process, and therefore in tumor invasion and metastasis, is vimentin [51]. Vimentin is one of the major intermediate filament proteins and is ubiquitously expressed in normal mesenchymal cells [80]. Recent studies have reported that vimentin knockdown causes a decrease in genes linked to HBC metastasis, such as the receptor tyrosine kinase Axl [81]. In our study, we also evaluated the expression of vimentin in HBCs, CMTs, and FMTs. We found a higher expression of vimentin in TNBCs compared with ER+, although not statistically significant. This result suggests that vimentin expression is associated with the triple-negative subtype, aggressive behavior, and a poor prognosis of HBC, as previously reported by many authors [82,83,84,85]. In CMTs, vimentin expression is low (approximately 15%), con-firming the low aggressiveness of mammary tumors in dogs, which is in concordance with the findings of other authors [86]. Conversely, in FMTs, the expression of vimentin, although heterogeneous, was quite high (approximately 70%), suggesting the high aggressiveness of mammary tumors in this species [9], as well as their similarities with TNBCs [11].

Unfortunately, as a limitation of this study, only grade I and II CMTs were included. No RNALater-sampled canine tumors were diagnosed as grade III. For possible IHC analyses in our archive of paraffin-embedded tissues, a very limited number of grade III simple CMTs were found (14 cases over five years) that were often already vascular/lymph node invasive (10/14). This study would not benefit much from adding only IHC analysis of grade III CMTs that already have invaded the vascular system or with metastases. We still believe that the study allowed the collection of some new data on the most frequent FMTs and CMTs in comparison with HBC samples assessing both gene and protein expression. 

## 5. Conclusions

In summary, this study showed that most of the classical EMT-related transcription factors *SNAIL*, *TWIST*, and *ZEB* are downregulated in tumor tissues compared with healthy tissues, although additional analyses should be performed to better investigate them in neoplastic clones and in a larger set of samples. IHC analyses indicated a potential role of some markers, namely vimentin and E-cadherin, but not of others (i.e., CD44) as indicators of EMT (including loss of cell differentiation and increased malignancies). Moreover, all the IHC data seem to support the already proposed similarities between FMTs (grade III) and TNBCs, as well as between CMTs (grade I and II) and ER+ HBCs. The two species are widely discussed as potential spontaneous models of specific HBC subtypes [11,12,15,16,57,87,88,89,90].

## Figures and Tables

**Figure 1 animals-13-00878-f001:**
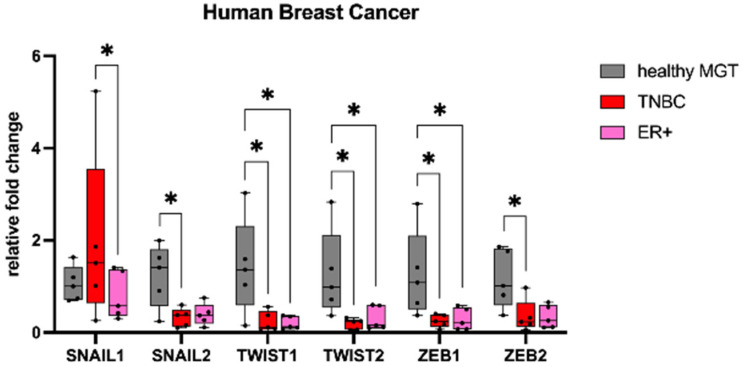
mRNA expression of *SNAIL1, SNAIL2, TWIST1, TWIST2, ZEB1*, and *ZEB2* in human breast cancer. *SNAIL1* showed a higher mRNA expression in TNBCs when compared with ER+ and healthy MGTs. *TWIST1, TWIST2*, and *ZEB1* mRNA expression was lower in estrogen receptor-positive (ER+) and triple-negative breast cancer (TNBCs) than in healthy tissues. *SNAIL2* and *ZEB2* mRNA expression was lower in TNBCs than in healthy tissues. MGT, mammary gland tissue. * *p* < 0.05.

**Figure 2 animals-13-00878-f002:**
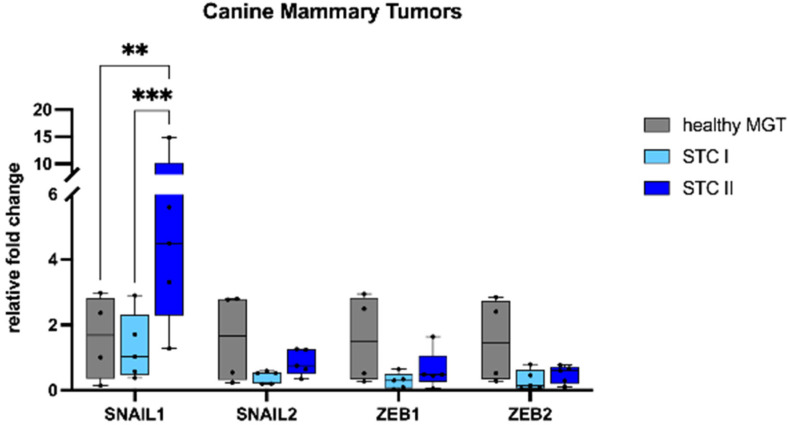
mRNA expression of *SNAIL1, SNAIL2, ZEB1*, and *ZEB2* in canine mammary tumors. *SNAIL1* mRNA expression was higher in simple tubular carcinomas grade II when compared with healthy tissues. *SNAIL2, ZEB1*, and *ZEB2* mRNA expression was lower in tumors when compared with healthy tissues. MGT, mammary gland tissue; STC I, simple tubular carcinoma grade I; STC II, simple tubular carcinoma grade II. ** *p* < 0.01, *** *p* < 0.001.

**Figure 3 animals-13-00878-f003:**
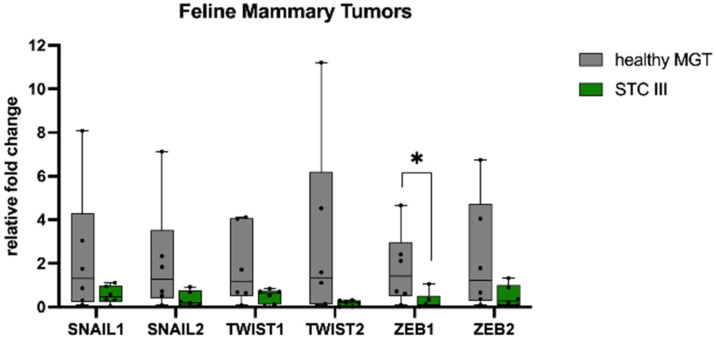
mRNA expression of *SNAIL1, SNAIL2, TWIST1, TWIST2, ZEB1*, and *ZEB2* in feline mammary tumors. *SNAIL1, SNAIL2, TWIST1, TWIST2, ZEB1*, and *ZEB2* mRNA expression was lower in simple tubular carcinomas (STC) grade III than in healthy tissues. MGT, mammary gland tissue. * *p* < 0.05.

**Figure 4 animals-13-00878-f004:**
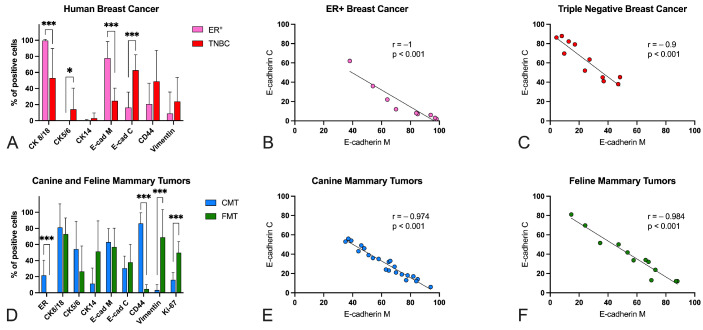
Immunohistochemistry results for cytokeratin (CK) 8/18, CK5/6, CK14, membranous E-cadherin (E-cad M), cytoplasmic E-cadherin (E-cad C), CD44, vimentin in estrogen receptor-positive (ER+) and in triple-negative breast cancers (TNBCs) (**A**). CK8/18 and membranous E-cadherin expression was lower in TNBCs than in ER+. CK5/6 and cytoplasmic E-cadherin expression was higher in TNBCs than in ER+. A strong negative correlation between membranous and cytoplasmic E-cadherin expression was found in ER+ (**B**) and in TNBCs (**C**). Immunohistochemistry results for ER, CK8/18, CK5/6, CK14, E-cad M, E-cad C, CD44, vimentin, and Ki-67 in canine (CMTs) and feline mammary tumors (FMTs) (**D**). ER and CD44 protein expression was lower in FMTs than in CMTs. Vimentin and Ki-67 expression was higher in FMTs than in CMTs. A strong negative correlation between membranous and cytoplasmic E-cadherin expression was found in CMTs (**E**) and FMTs (**F**). * *p* < 0.05; *** *p* < 0.001.

**Figure 5 animals-13-00878-f005:**
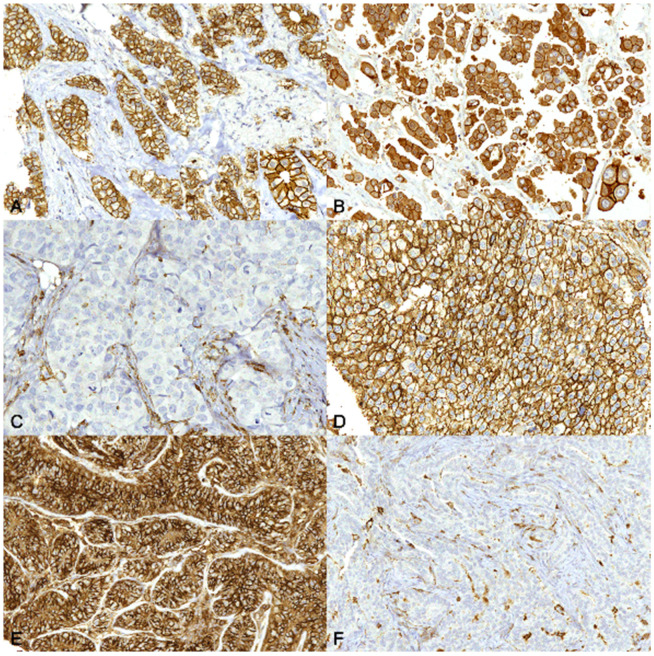
Carcinoma, mammary gland. Immunohistochemistry for E-cadherin (**A**,**B**) and CD44 (**C**–**F**). 20× magnification. (**A**) Human. Invasive ductal carcinoma, grade I. In estrogen receptor-positive (ER+) human breast cancer, immunolabeling for E-cadherin is mainly membranous. Inset: higher magnification. (**B**) Human. Invasive ductal carcinoma, grade III. In triple-negative breast cancer (TNBC), immunolabeling for E-cadherin is sometimes partially lost from the cell membrane and often present within the cytoplasm. Inset: higher magnification. (**C**) Human. Invasive ductal carcinoma, grade III. CD44 expression is rarely positive in ER+ tumors. (**D**) Human. Invasive ductal carcinoma, grade III. Membranous CD44 immunolabeling is diffusely expressed in TNBCs. I Canine. Simple carcinoma grade II. The immunolabeling for CD44 is membranous and diffusely evident. (**F**) Feline. Simple carcinoma, grade III. CD44 expression is negative in tumor cells.

**Figure 6 animals-13-00878-f006:**
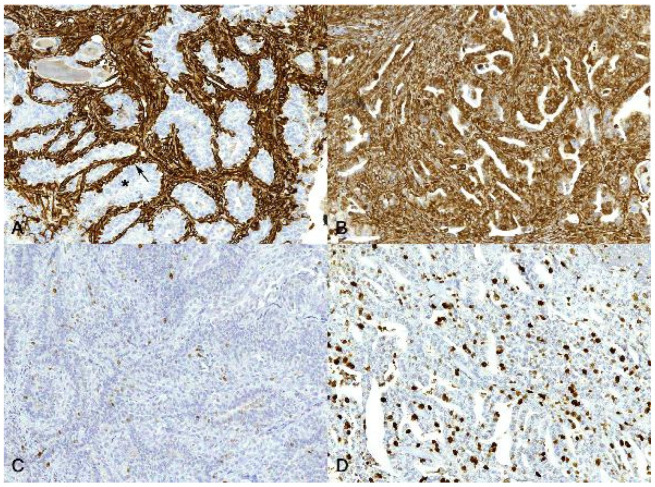
Carcinoma, mammary gland. Immunohistochemistry for vimentin (**A**,**B**) and Ki-67 (**C**,**D**). 20× magnification. (**A**) Canine. Simple carcinoma, grade II. In canine mammary tumors (CMTs), vimentin expression is negative in luminal cells (asterisk) and positive in basal/myoepithelial cells (arrow). (**B**) Feline. Simple carcinoma, grade III. In feline mammary tumors (FMTs), vimentin expression is evident in approximately 90% of tumor cells, including luminal cells. (**C**) Canine. Simple carcinoma, grade I. In CMTs, Ki-67 expression is detectable in approximately 10% of tumor cells. (**D**) Feline. Simple carcinoma, grade III. In FMTs, Ki-67 expression is evident in approximately 40% of tumor cells.

**Table 1 animals-13-00878-t001:** Primer sequences used in this study.

	HUMAN	DOG	CAT
SNAIL1	F: 5′-TTCTCACTGCCATGGAATTCC-3′ R: 5′-GCAGAGGACACAGAACCAGAAA-3′	F: 5′-ACTGCAGCCGTGCCTTTG-3′ R: 5′-AAGGTTCGGGAACAGGTCTTG-3′	F: 5′-CACCTGTTTCATGGGCAATTT-3′ R: 5′-CATCGGTCAGGCTGAAGCA-3′
SNAIL2	F: 5′-GCACACTGAGTGACGCAATCA-3′ R: 5′-AGCACAGGAGAAAATGCCTTTG-3′	F: 5′-TTTTCTGGGCTGGCCAAA-3′ R: 5′-CGCCCAGGCTCACGTATT-3′	F: 5′-TGCAGACCCATTCGGATGT-3′ R: 5′-CAGCAGCCAGATTCCTCATGT-3′
TWIST1	F: 5′-GTCTAGAGACTCTGGAGCTGGATAACT-3′ R: 5′-CGCCCTGTTTCTTTGAATTTG-3′	-	F: 5′-TTAGAAGAGCAGAACCCAAAT-3′ R: 5′-CTGCCCGTCTGGGAATCA-3′
TWIST2	F: 5′-AGGACGGTCCCCACATAGG-3′ R: 5′-ACATAAGACCCAGAAGAAAAATCCA-3′	F: 5′-CAGACACGGTCCCCACACA-3′ R: 5′-AACCCAGAAGAAAAGATCCAAACA-3′	F: 5′-GGAAACGCGACGCTGAGT-3′ R: 5′-GGAAGCCACAGATGCACTTTG-3′
ZEB1	F: 5′-GATGATGAATGCGAGTCAGATGC-3′ R: 5′-ACAGCAGTGTCTTGTTGTTGT-3′	F: 5′-AAAATGAGCAAAACCATGATCCTAA-3′ R: 5′-CCCTGCCTCTGGTCCTCTTC-3′	F: 5′-CCCACACGACCACAGATAAGG-3′ R: 5′-TGAATTCATAATCCACAGGTTCA-3′
ZEB2	F: 5′-CCAGCTCGAGCGGCATA-3′ R: 5′-GCCACACTCTGTGCATTTGAA-3′	F: 5′-TTACCCAGGTCGCCCGTAA-3′ R: 5′-TTAGCCTGAGCGGAGGATCA-3′	F: 5′-CACGATCCAGACCGCAGTTA-3′ R: 5′-GTCGCGTTCCTCCAGTTTTC-3′
ACTB	F: 5′-TGGCACCACACCTTCTACAA-3′ R: 5′-CCAGAGGCGTACAGGGATAG-3′	F: 5′-TGGCACCACACCTTCTACAA-3′ R: 5′-CCAGAGGCGTACAGGGATAG-3′	F: 5′-TGGCACCACACCTTCTACAA-3′ R: 5′-CCAGAGGCGTACAGGGATAG-3′

**Table 2 animals-13-00878-t002:** Immunohistochemistry results in human (HBC), canine (CMT), and feline (FMT) mammary cancer represented as mean ± standard deviation (SD) of percentage of positive cells. Statistics refers to the comparison of the expression of a specific marker between groups of the same species.

IHC (Mean ± SD)	HBC ER^+^	HBC TNBC	CMT STC I	CMT STC II	FMT STC III
*ER*	81.7 ± 18.4	negative	30.4 ± 17.4 **	13.1 ± 16.4 **	negative
*PR*	57.5 ± 42.6	negative	NP	NP	negative
*ERBB2*	0 or 1+ ^#^	negative	negative	negative	negative
*CK8/18*	99.5 ± 1.6 ***	53.0 ± 36.9 ***	91.7 ± 10.4	70.8 ± 38.0	73.0 ± 19.9
*CK5/6*	Negative *	14.2 ± 26.22 *	54.9 ± 32.9	53.6 ± 37.5	26.4 ± 31.8
*CK14*	0.4 ± 0.8	3.2 ± 6.3	9.6 ± 11.7	13.1 ± 25.4	51.3 ± 37.9
*CD44*	20.7 ± 25.8	48.9 ± 38.6	85.7 ± 14.5	86.8 ± 12.4	4.7 ± 5.4
*E-cad M*	77.8 ± 20.5 ***	24.7 ± 15.7 ***	58.9 ± 18.8	66.9 ± 14.8	57.0 ± 23.3
*E-cad C*	16.4 ± 19.1 ***	62.7 ± 19.3 ***	35.1 ± 15.7	25.8 ± 13.6	37.9 ± 22.3
*Vimentin*	8.9 ± 26.7	23.8 ± 30.1	4.2 ± 9.3	2.5 ± 3.1	68.9 ± 34.3
*Ki-67*	NP	NP	14.0 ± 7.0	18.4 ± 10.4	49.6 ± 13.9

ER, estrogen receptor; PR, progesterone receptor; E-cad, E-cadherin; M, membrane; C, cytoplasm; TNBC, triple-negative breast cancer; STC I, simple tubular carcinoma grade I; STC II, simple tubular carcinoma grade II; STC III, simple tubular carcinoma grade III. ^#^, ERBB2 was scored as 0, 1+, 2+, and 3+ according to the American Society of Clinical Oncology (ASCO) 2018 recommendations (10% cut-off). None of the samples was scored as 2+ or 3+. * *p* < 0.05; ** *p* < 0.01; *** *p* < 0.001. NP, not performed.

## Data Availability

The data presented in this study are available upon request from the corresponding author.

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
