# Peer review of "Epithelial-to-Mesenchymal Transition and Phenotypic Marker Evaluation in Human, Canine, and Feline Mammary Gland Tumors"

_animals, 2023, doi:10.3390/ani13050878_

Round 1

Reviewer 1 Report

The paper is very interesting and brings new information concerning EMT in human, canine and feline mammary tumor. 

The images are very beautiful, especially the immunohistochemistry.

I suggest adding some results comparing the gene expression profile of the tumors, with the protein expression. For example: in feline mammary tumor, does ZEB1 lower expression, correlate with an EMT protein expression? Higher vimentin expression? Higher cytoplasmic E-cad expression?

In lines 354/362: SNAIL1 is higher in CMT, and how about E-cad expression? Was it lower?

Please, use italic for gene names, when referring to mRNA expression.

Methods:

It is not clear (lines 106/108) if the tissue that was formalin fixed was the same as it was preserved in RNA later. I believe that the researchers collected one sample for RNA later and mRNA extraction and another one FFPE. Is that correct? This should be better explained. If the sample was preserved in RNA later and then fixed in paraffin, what was the point of the RNA later?

How many samples were collected from each tumor?

For FFPE and mRNA extraction, did you collect from the same area?

Please, be more specific of the primers sequence. How did you design the primers? Did you use any program or a reference from a paper?

Were the primers sequencies the same for all species (human, canine and feline)?

Figure 1: Why there is only * to show difference in SNAIL1 expression between TNBC and ER+ tumors, and not with MGTs, if in the text it is written that that was a statistical difference? (lines 178/179).

Why did you not add TWIST expression in canine results (figure 2), and for human and feline TWIST1 and TWIST2 are presented, although with no statistical difference?

Figure 5: there are no insets (as in the legend) (line 318, 321). There is no arrow in 5C.

Figure 6 – legend: what did you mean by moderate number of cells. In methods the number of positive cells were counted and a % made. So, moderate correlates to what % of positive cells?

Lines 420/421 – how did you compare E-cad expression with tumor aggressiveness? This information is not presented in results (concerning tumor aggressiveness). 

Author Response

thank you very much to the reviewer, please find our feedback int he attached file 

Reviewer 2 Report

In this paper, the authors studied EMT related genes and phenotypic marker in mammary gland tumors from three different species. The author’s manuscript is potentially intersting. Despite this premise, there are several concerns regarding this study and the authors need to be clarify. 

-       The number of tissue samples is low. I strongly suggest to the authors to add grade 3 (High-grade) canine mammary gland tumors (CMT) in cohort. Grade 2 CMTs behave similarly to grade 1 CMTS, with prolonged overall survival times (Rasotto et al. Prognostic Significance of Canine Mammary Tumor Histologic Subtypes: An Observational Cohort Study of 229 Cases. Veterinary Pathology. 2017;54(4):571-578). Inclusion of Grade 3 could provide a broader understanding of the pattern of CMT.

-       None of the CMT samples was ERBB2 positive according to authors’ results. Did you classify it as negative by judging cytoplasmic expression as a background based on the 2018 standard? Or was there really no expression of more than 10% in the membrane? In addition to ERBB2, please provide other IHC results such as ER and PR as supplementary data.

Author Response

thank you to the reviewer for the comments, please find attached our feedback

Round 2

Reviewer 2 Report

It is unfortunate that there are no Grade 3 CMT samples obtained with RNAlater. And I understand the challenges of obtaining samples. I agree with your opinion that adding IHC analysis using grade 3 CMT samples would not add much value unless EMT-related gene analysis in grade 3 CMT samples. In the discussion section, please describe as a limitation that only Grade 1 and Grade 2, which show similar tendencies, were included in this research and Grade 3 CMT samples, which show aggressive tendencies, were not included.

Nevertheless, I think this paper is valuable because authors analyzed and compared EMT related genes in human, canine and feline samples.

Author Response

thank you very much for you rapid response and comment. we added a sentence at the end of the discussion to include the limitation regarding CMTs. We really appreciated your understanding of the difficulty of collecting homogeneous samples in veterinary oncopathology study.